# What Room for Two-Dimensional Gel-Based Proteomics in a Shotgun Proteomics World?

**DOI:** 10.3390/proteomes8030017

**Published:** 2020-08-06

**Authors:** Katrin Marcus, Cécile Lelong, Thierry Rabilloud

**Affiliations:** 1Medizinisches Proteom-Center, Medical Faculty & Medical Proteome Analysis, Center for Proteindiagnostics (PRODI) Ruhr-University Bochum Gesundheitscampus, 4 44801 Bochum, Germany; katrin.marcus@rub.de; 2CBM UMR CNRS5249, Université Grenoble Alpes, CEA, CNRS, 17 rue des Martyrs, CEDEX 9, 38054 Grenoble, France; cecile.lelong@univ-grenoble-alpes.fr; 3Laboratory of Chemistry and Biology of Metals, UMR 5249, Université Grenoble Alpes, CNRS, 38054 Grenoble, France

**Keywords:** two-dimensional gel electrophoresis, proteomics, post-translational modifications, proteoforms, clinical proteomics

## Abstract

Two-dimensional gel electrophoresis was instrumental in the birth of proteomics in the late 1980s. However, it is now often considered as an outdated technique for proteomics—a thing of the past. Although this opinion may be true for some biological questions, e.g., when analysis depth is of critical importance, for many others, two-dimensional gel electrophoresis-based proteomics still has a lot to offer. This is because of its robustness, its ability to separate proteoforms, and its easy interface with many powerful biochemistry techniques (including western blotting). This paper reviews where and why two-dimensional gel electrophoresis-based proteomics can still be profitably used. It emerges that, rather than being a thing of the past, two-dimensional gel electrophoresis-based proteomics is still highly valuable for many studies. Thus, its use cannot be dismissed on simple fashion arguments and, as usual, in science, the tree is to be judged by the fruit.

## 1. Introduction

A decade after its introduction in the mid-1970s [1,2], two-dimensional electrophoresis was instrumental in the birth of proteomics [3] Since then, the exponential development of shotgun proteomics, pioneered at the turn of the century [4] and driven in great part by the gigantic increase in the sensitivity of mass spectrometers, has largely superseded two-dimensional (2D) gel-based proteomics. However, although often being referred to as an outdated method, 2D gel-based proteomics still has some merit, as recently reviewed [5,6,7]. The purpose of this review is to expand on this topic and emphasize the positive features that make 2D gel-based proteomics an option to consider in some circumstances. 

First of all, the analysis depth is a question to consider, as 2D gel-based proteomics is often criticized on this ground. Modern 2D gels visualized with sensitive techniques (e.g., differential in gel electrophoresis (DIGE) or silver staining) often reach 2500–3000 individual spots. With an average of three spots per protein in eukaryotes [8], this corresponds to ca. 800 gene products. According to shotgun data, 800 proteins correspond themselves to 90% of the cellular protein mass [9], i.e., 90% of the energy invested in the cell in producing proteins. Thus, as a cost function, 2D gel-based proteomics is still a relevant approach, especially when the mechanistic details of the cellular responses, i.e., unveiling in detail the signaling pathways or networks at play, are not the core of the biological question.

Concisely, 2D gel electrophoresis is the only currently available technique that is able to separate complete proteins over a wide pI (isoelectric point) and molecular mass range, with a resolution high enough so that a spot corresponds most often to one major protein. With the onset of high sensitivity mass spectrometry (MS), every spot observed in a 2D map now leads to the identification of several proteins in it. This has been perceived as a concern of variable importance [10,11]. Recent research, carried out in the worst possible setup in terms of multiple identifications (i.e., coupling a high sensitivity MS to highly loaded 2D gels, where the Gaussian spreading of spots and the presence of streaks were maximized) has shown that the most abundant protein in a spot, as detected by MS, most often accounts for >75% of the total signal [12]. Although a previous paper had described that, in 30% of the cases, the most abundant protein accounted for less than 70% of the “protein intensity” [13]. It should be emphasized, however, that such calculations rely on MS-based indexes such as emPAI (exponentially modified protein abundance index), which have been shown to lack accuracy [14,15,16]. This shall not come as a surprise owing to the variations in peptide to mass signal yield, as detailed in the next section.

Furthermore, the corpus literature on identification of proteins from 2D gel spots by the peptide mass fingerprinting approach (reviewed in [17]) further exemplifies that unambiguous identifications are more the rule than the exception in two-dimensional gel electrophoresis (2DGE) proteomics.

Moreover, should a doubt persist, a third separation can always attempt to decipher, more precisely, the protein content of a spot of interest [18,19].

This ability to analyze complete proteins, with a view on their post-translational modifications, is of high theoretical and practical interest in proteomics. In an ideal world, to maximize the biological relevance of the method, proteomics should be able to analyze complete, native, folded proteins with their cohort of post-translational modifications and prosthetic groups. As a matter of fact, there are examples where the simple alteration of a loosely bound prosthetic group, such as an iron-sulfur cluster, can completely change the protein function at constant polypeptide chain [20].

This is unfortunately impossible with current technology, so that only a proxy to the real biological object of interest can be analyzed. In this context, a denatured protein, with its cohort of covalent post-translational modifications, is the best possible proxy, before the pool of all modified forms of the same protein (i.e., the level of the gene product), preceding itself digestion peptides produced from the proteins.

This being stated, the question that arises is to determine to which extent this theoretical advantage tells, knowing that 2D gel-based proteomics has a narrower analysis window than shotgun proteomics in terms of pI, MW (molecular weight), and protein hydrophobicity, is more sample consuming and slower to carry out. The purpose of this review is to investigate this question.

## 2. Peptides or Proteins—That Is the Question

Most MS methods in proteomics involve the analysis of peptides instead of proteins. This strategy is generally referred to as “bottom-up” analysis [21]. In this approach, the proteins are cleaved enzymatically or chemically into peptides, the mixture is analyzed by MS, then identified in a database search and, if required, quantified and then reclustered into “proteins” (in fact, on a gene product scale) in silico. In contrast, “top-down” proteomics is used to analyze proteins as a whole, as in the 2D-PAGE-based approaches or in newer MS-based approaches [22,23] (a comparison of the two workflows is shown in Figure 1). These strategies make it possible to obtain more complete information on proteoforms, which can be very relevant depending on the scientific question. The term “proteoform” describes all different molecular forms of a protein product of a single gene, including changes due to genetic variations, alternatively spliced RNA transcripts, and post-translational modifications [24].

Bottom-up strategies have evolved from the situation that it is much easier to analyze peptides by MS in high-throughput than proteins. Peptides have more similar physicochemical properties, such as solubility, hydrophobicity, and separation behavior among each other than proteins. After digestion and mass spectrometric measurement, characteristic peptide patterns are obtained for each protein, which are referred to as peptide mass fingerprints [25].

In tandem MS approaches, peptide ion mass spectra (MS1 spectra) are first generated. The peptide ions are then further fragmented inside the mass spectrometer, resulting in fragment ions mass spectra (MS/MS or MS2 spectra). In this way, more specific information is obtained for each peptide and a better identification of the peptides is possible [26]. The generated data are analyzed using database search algorithms, such as SEQUEST [27], Mascot [28], X! Tandem [29], etc., and further processed by analysis programs, such as MaxQuant [30], Proteome Discoverer [Thermo Scientific], or Progenesis QI [Nonlinear]. The results are then peptide spectrum matches (PSMs). In a next step, the obtained PSM data are reclustered into a gene product. This process is called protein inference [31].

Brain T. Chait writes in his 2006 article in *Science* that “Unfortunately, only a small fraction of tryptic peptides are normally detected…. The bottom-up approach is therefore suboptimal for determining modifications and alternative splice variants” [21] (i.e., proteoforms). This means that by analyzing the peptides, we only obtain a section of all information on the associated proteins and important information on proteoforms is lost.

Apart from the fact that not all peptides of a protein are measured and identified, and the protein information is, therefore, only fragmentary, there is an additional challenge in data analysis: it might not be possible to clearly determine which of the proteoforms present in the sample a peptide can be assigned. The bottom-up approach specifies that the measured peptide data are used to assign to a specific protein. However, a clear assignment between peptide and protein is only possible if the detected peptide is a unique peptide (i.e., if it is a peptide that only occurs in a single protein and is, therefore, clearly specific to that protein at a given time of knowledge). A significant number of peptides are not unique but shared by different proteins in the database, especially in eukaryotic organisms [32]. These shared peptides lead to sets of proteins (protein ambiguity groups), which are created, from the same (sub) set of measured peptides. Finally, without any unique peptide evidence, it cannot be determined, which of the proteins of an ambiguity group was/were actually present in the original sample. Some of the database search algorithms and programs try to solve this issue by using only unique peptides for inference or reporting protein groups or representatives as a result. There have been some developments in recent years that address the problem of protein inference [33,34,35,36,37,38]. It is important for us to emphasize this point again here and to draw the reader’s attention to the fact that the MS and MS/MS data evaluated may need to be viewed critically.

In addition, bottom-up analyses present additional challenges for label-free quantitative (LFQ) proteome analysis. The search algorithms or programs use different strategies to deduce the protein quantity from the peptide quantity. Ultimately, however, the aim is to draw conclusions from the measured intensities of the peptides about the intensity and hence the quantity of the protein and its variants. This means that peptides are regarded as representatives for the proteins. If not only unique peptides are included in this calculation, then there is a risk of incorrect quantification, since the intensities (and thus also quantities) of the shared peptides reflect those of several proteins/proteoforms. However, if only unique peptides are included in the quantitative analysis, of which there may be only a few, then the quantification of the protein may rely on a less reliable data basis. Other strategies approaching this problem use, e.g., the covariation of peptides’ abundances in all samples [39].

We would like to illustrate the problem with a concrete example. In a cerebrospinal fluid (CSF) study [40], which we partially published in 2018, and which aimed to evaluate potential published biomarkers for Parkinson’s disease (PD); we examined the protein “haptoglobin” (Hp). For Hp, two isoforms are described, where isoform 2 differs from the canonical sequence in that amino acids 38–96 are missing. In addition, various glycosylations and disulfide bridges are known, i.e., other proteoforms exist.

Even without consideration for its isoforms, this protein is described as a potential protein biomarker candidate for several neurological diseases such as PD [41,42,43], Alzheimer’s [44], multiple sclerosis [45], hypertrophic cardiomyopathy [46], as well as ovarian cancer [47,48], and many others. This protein had been described as a potential biomarker for PD, but different studies showed a different tendency with regard to regulation in CSF and/or serum [49]. We generally found a very high variability for the Hp in CSF. Our LFQ analysis also revealed that of the 21 peptides assigned to Hp (12 of them unique), some behaved very differently in terms of intensity within a sample (see Figure 2). From these and other data from our projects, which we have critically evaluated again afterwards, we have concluded that it is not always possible to draw clear conclusions about the amount of protein from the amount of peptides. Quantification at peptide level alone might be a possible solution to this problem. In this way, the protein inference step, which can lead to non-uniform assignments as described above, is avoided.

When looking at the Hp spot pattern in a 2D gel, we find that it is distributed over a large number of different spots across the gel [https://world-2dpage.expasy.org/swiss-2dpage/protein/ac=P00738]. A 2DGE-based study of ovarian cancer [47] clearly detected seven potentially disease-relevant well-separated Hp proteoforms (including modified alpha 1 and 2 isoforms) in ascitic fluid of patients. Authors found an association between the number of alpha isoforms and the disease stage of patients. Previously, several studies have already identified Hp and particularly the fucosylated forms in serum of patients and suggested its possible use as a diagnostic biomarker [50,51]. Indeed, only a top-down approach using 2DGE in combination with a fucosylation specific lectin assay allowed for the differential detection/identification of proteoforms, especially alpha subunit expression together with differential levels of fucosylation.

By contrast, and by design indeed, 2DGE proteomics is free from these issues. In this proteomic setup, the quantification step is not carried out by MS, but through the 2D gel images. As proteins and not peptides are separated in 2D gels, proteins are directly quantified on the images. Opposite to peptides, which show a high variability in their detectability and response factor (as exemplified by the different signals observed for different peptides arising from the same protein, which should be equimolecular except for shared peptides), there is an averaging/buffering effect in proteins that makes their quantification less variable from one protein to another.

Thus, besides the identification issues discussed earlier [10,11,12], the quantification issues in 2DGE proteomics lie mainly with the performances of the detection methods used, both in terms of sensitivity and linearity.

Regarding linearity, detection by fluorescence is clearly the best option, as fluorescence can be linear over several orders of magnitude, which is typically what proteomic analyses face. One of the most popular setups for fluorescence detection is the so-called DIGE setup (reviewed in [52]), but there are many different options for fluorescent detection and quantification of proteins after 2DGE, using covalent labelling of proteins or not [53], including detection of the classical Coomassie blue by fluorescence [54]. At equal dye, the comparison of the performances achieved by fluorescence [54] to those achieved by light absorption [55] clearly shows the power of fluorescent detection. Sensitive fluorescence detection requires however expensive hardware, so that classical detection by light absorption, which can be achieved on a classical scanner, is often favored. In this setup, organic dyes are plagued by a relatively low sensitivity [55], and are superseded in this respect by silver staining [56]. Silver staining however suffers from a low response curve [56], so that the quantitative differences highlighted by silver-stained 2D gels may be an under-representation of the true quantitative changes at the protein levels. This further emphasizes the fact that arbitrary thresholds should be avoided for quantitative proteomic analyses [57].

Incidentally, this combination of high sensitivity detection and homogeneous response of proteins makes 2DGE proteomics an attractive choice for a niche application in which the aim is not to compare different samples, but to quantitatively analyze one sample for its different proteins. One example of this case is represented by the analysis of therapeutic protein batches, where 2DGE has been successfully applied [58,59]. Indeed 2DGE is very good at detecting not only the predictable, such as contaminant host cell proteins, which can be analyzed by shotgun proteomics [60], but also the more difficultly-predictable, such as modifications of the therapeutic protein of interest, which can translate into the appearance of new protein spots.

In summary, it can be said that the current bottom-up strategies have their advantages and disadvantages. Even though it is easier to analyze peptides instead of proteins, the subsequent qualitative or quantitative data analysis must be critically reviewed and should ideally be inspected manually in order to avoid betting on the wrong horse in a subsequent extensive evaluation of the results. Indeed, important systemic technical biases have been documented for shotgun proteomics [61]. Conversely, the reliability of 2DGE proteomics, in this respect, is an often underlooked and underestimated advantage of the setup.

## 3. When an Unpredictable Subset of Proteins Is to Be Analyzed: The Example of Immunome/Allergome Studies

The first area in which 2D gel-based proteomics shows interesting performances are the studies where the point of interest is to decipher which proteins of a pathogen/allergen are targeted by the immune system of the host. The great advantage of 2D gel-based proteomics in this type of research is that it uses the host antibodies as an analytical reagent to detect the proteins of interest, and not as a preparative reagent to select the proteins of interest. Using antibodies as preparative reagents is associated with many problems associated with the constraints brought by the solid supports that must be used, which bind a wide variety of proteins, leading to a very extensive and severe background [62,63]. These problems do not happen when antibodies are used as analytical reagents because of the tricks (e.g., saturation with other proteins) that can be applied in this scheme. The combination of 2D gels to display the target organism proteins and 2D blots to highlight the proteins recognized by the host immune system, looping back to the target organism protein 2D gels to identify then the proteins of interest, has proven very efficient in many cases. Although theoretically straightforward, this process is practically not as easy as it may seem, as it requires a very rigorous matching of the 2D blot pattern on the gel pattern to avoid any mistake at this stage. This has led to the development of several methods, from the detection of the total protein pattern on the 2D blots prior and/or after the immunodetection (e.g., in [64]) to the partial blotting process, in which the very same 2D gel is used, both for generating the 2D reference pattern for MS identification and the 2D blot for immunodetection [65]. In the immune responses to pathogens, work has been carried out on bacteria [66,67,68,69,70], fungi [71,72,73], and parasites [74,75]. Such studies have always reached their initial goal of providing the major immunogens from the pathogen under study, and have sometimes resulted in valuable clinical advances [71,76,77].

The situation is very similar in the allergy field, where as a further refinement only the IgEs (immunoglobulin E) of the patients are selectively detected. The response to various allergens has been studied, as exemplified recently in [78,79,80,81], and reviewed earlier in [82].

Lastly, this approach has also been used in diseases with an autoimmunity component to identify autoantigens in various pathological contexts such as arthritis (e.g., in [83,84,85,86,87]), multiple sclerosis (e.g., in [88,89]), or other diseases [90,91].

These examples show the convenience and interest of 2D gel-based proteomics in the immunome/allergome field, where experimental schemes based on shotgun proteomics are plagued by the issues linked to antibody immobilization mentioned above.

## 4. Going to the Essence of Proteomics: Proteoforms and Post-Translational Modifications

If comprehensiveness is defined from a genetic/genomic standpoint, i.e., from the 1941 one gene-one protein dogma [92] and, thus, from the number of gene products identified, proteomics lacks comprehensiveness, even in its deepest shotgun versions, compared to transcriptomics, which has reached comprehensiveness through the use of deep sequencing [93]. Thus, the value of proteomics lies in the fact that it analyzes a better proxy of the cellular functions, i.e., proteins. In doing so, all proteomics setups integrate the regulations that take place at the translational level. However, it appears more and more shiningly that a great deal of regulations occur at the post-translational level, mostly through post-translational modifications (PTMs). There are literally dozens of different modifications, which are enzyme catalyzed for some of them but sometimes not [94,95,96]. Among the enzyme-catalyzed modifications, phosphorylation is the one described for the longest time and which role is best known. Glycosylation is also known for a very long time, but its daunting complexity [97] has made progress slower in understanding the full scope of its functions. More recently other modifications such as methylation [98,99], acetylation [100,101,102,103], or other acylations [104,105,106] have been described and modulate protein localization and/or function.

This increased recognition of the importance of PTMs has led to the concept of proteoform(s), i.e., a protein backbone bearing a precise combination of PTMs. Ideally, proteomics should focus on the study of proteoforms and not on gene products. This is easier said than done, owing to the number of technical and operational challenges that arise when studying proteoforms. In this context, any tool that is able to separate a protein into a subset of proteoforms even if this separation is not complete, is a step further in moving proteomics in this direction. Although some PTMs are electrophoretically neutral (e.g., Lys/Arg methylation, Cys acylation) many are not, such as Lys acylation or phosphorylation. Because of its resolution in the isoelectric focusing (IEF) dimension, which is where the electrophoretic impact of PTM is most easily seen, 2D electrophoresis is, therefore, able to separate many protein variants on the basis of their pI. Consequently, most proteins appear not as a single spot, but as a trail of spots, with an average of 3 spots/protein in mammalian cellular proteins [8]. This relatively low number shall not be taken as an average number of proteoforms, as IEF just counts the number of modifications but gives no information on their localization. For example, proteoforms bearing the same number of phosphorylations, but at different positions, will be merged into a single spot. In fact, full characterization of discrete protein spots have shown that a single spot can be a mixture of differently-modified proteins [107].

Despite this lack of resolving power when compared to ideality, the separation afforded by 2DGE is clearly an advantage when trying to bring proteomics closer to the proteoform world [5,6,7]. Consequently, there are literally thousands of scientific articles that bring together the keywords “2D electrophoresis” and “post-translational modifications”, and this article will of course not aim at citing all of them. It will rather cherry pick some papers that appear of interest in the dimensions selected and highlighted here.

### 4.1. Guidelines for Use

The advantage of being able to resolve proteins as several discrete spots brings its own set of challenges and issues that must be taken into account when using 2DGE in proteomics. With an average of three spots/protein [8], when 2DGE is used in differential proteomics, the most frequent situation is that only one (or a few) protein spots will change under the biological condition of interest, but not all the spots corresponding to the same protein. Murphy’s Law being what it is, the modulated spots are generally the weakest and the most modified, i.e., the farthest from the theoretical position of the protein. There is thus a danger to mistake the part for the whole, and to overclaim “the amount of protein X is changed” while the true situation is “the amount of form Y of protein X is changed”. In order to get a good appraisal of the real situation, it is, therefore, highly advisable to try to look for other forms of the protein of interest in the 2D gel space. This is more or less easy to do depending on the chemical characteristics of the proteins of interest, leading to the notion of separation cone in 2D electrophoresis. Indeed, depending on the number of charged amino acids that they contain, proteins are more or less “buffered” in the IEF dimension. Thus, the change brought by the same PTM will result in a different spacing in the IEF dimension depending on the protein. As an example, a high molecular weight protein such as matrin 3 contains many charged amino acids. Consequently, a PTM will lead to a minor spot spacing, so that the modified forms of the protein will appear as an easily recognizable train of spots [108]. Conversely, a lower molecular weight protein such as triosephosphate isomerase contains a lower number of charged amino acids, so that a single modification brings a much larger spacing on the 2D gels [109], which is much more difficult to take into account. Thus, as a first approximation, the spacing induced by PTM decreases when the molecular weight of the proteins increases, leading to the concept of separation cone. As always in the protein world, the situation is more complex, as proteins of the same molecular weight can have different charge densities and thus lead to a different spacing. Thus, a cautious and reasonable attitude when using 2DGE-based proteomics is, when an interesting spot is found, to try to figure out where the un-modulated major spots accounting for the bulk of the protein are, and to discuss the results accordingly.

### 4.2. Hypothesis Validation: Getting the Most from the 2DGE Data

Thus, the usual outcome of a 2DGE proteomic experiment is a list of modulated spots, and most often, the total amount of the protein of interest is not modulated in the experiment. This shall not mean the end of the story, as this situation is precisely what can be expected from a landscape where PTM play a major role in the modulation of protein activities, and this can be recognized upfront in very different perspectives, ranging from the description of modified protein forms in a therapeutic protein (e.g., in [59]) to functional differences (e.g., in [110]). However, the impact of PTM cannot be predicted a priori. Both for phosphorylation and for acetylation, sometimes the modifications increase the activity of the protein and sometimes it decreases it. Moreover, the classical LC-MS/MS (liquid chromatography tandem mass spectrometry) analysis of the spots carried out to identify the proteins of interest usually does not lead to the identification of the PTMs. In many cases, this is due to the fact that modifications are excluded from the analysis of the MS/MS spectra to limit the search space and the probability of false positives.

The absence of knowledge of the modification does not mean that the result is without biological interest. It just means that it shall be validated, directly or indirectly. In this respect, proteomic results are all equal, regardless of the setup used to produce them. By looking at quantitative changes, proteomics just exploit the homeostasis principle. As the composition of living systems tends to be constant, then if a perturbation of the system brings changes in the composition of the system, then these changes are relevant and important in the response to the perturbation. This ideal statement is tarnished by the fact that, as for all large-scale techniques, proteomics comes with its share of false positives and artifacts, so that a proteomic screen should be seen mostly as a hypothesis generator, and hypotheses must be validated.

This is most often a weak point of many proteomic papers, where the validation is kept to a minimum, to say the least, and most often consists of real-time quantitative polymerase chain reaction (RT-qPCR) and/or western blotting. This represents in a sense a kind of circular validation, in which a change of abundance detected by proteomics is just confirmed as being a change of abundance by another technique, without going much further in terms of biochemical knowledge. This may be due to the fact that validation is often research-intensive, and this poses in turn the problem of confidence in the proteomic results for validation.

One theoretical solution to this confidence problem would be to perform the proteomic analysis by two different techniques with different biases, and to consider what is found in common by both techniques as reliable results. This approach unfortunately does not apply to the combination 2DGE proteomics/shotgun proteomics. If the biases are different indeed, the two techniques target different analytes so that the results are not really comparable. There are, therefore, very few articles in which both approaches are used on the same samples, but these few are worth analyzing in detail.

In their work on schizophrenia [111], Martins de Souza et al. used both shotgun proteomics and 2DGE proteomics. As mentioned in their results, nine out of the ten differences that they found in 2DGE proteomics were not found by the shotgun proteomics screen. This should not come as a surprise in the light of the PTM effects described above. The reverse question is however worth being investigated. It is often stated that shotgun proteomics describes variations that are missed by 2DGE proteomics because of the greater analysis depth of shotgun proteomics. While this is certainly true in many cases, this study strongly suggests that it is not always the case. For example, phosphoglycerate mutase and transketolase are found modulated in schizophrenia by the shotgun proteomic screen, but not in 2DGE. These proteins belonging to the glycolysis and pentose phosphate pathways, respectively, they should be abundant proteins and detected easily in 2DGE, which they are indeed in other mammalian systems [112]. The fact that they were not detected in the 2DGE screen in the Martins de Souza et al. study [111] just means that they were not detected as variable, as 2DGE proteomics investigates only the proteins that change, and not all the detectable proteins, as shotgun proteomics does. There is thus an important discrepancy between the shotgun and 2DGE proteomics screen in the Martins de Souza et al. study concerning these two proteins, which are either a false positive of the shotgun proteomics or a false negative of the 2DGE proteomics. In the absence of enzyme activity data in the paper it is impossible to know which is correct.

Enzymes are indeed a very good illustrative case. They are regulated by modifications [100,113], so that they can be perceived as a good model for proteins in general, and the validation is often easy via the enzymatic activity. It can be argued that enzymes often belong to the “déjà vu in proteomics” [114], but this just points out their importance in stress response [115]. Indeed, the fact that the response is generic does not mean that it is unimportant. In this respect, it has been shown that preventing the induction of glyceraldehyde 3-phosphate dehydrogenase (GAPD), the epitome of a “déjà vu” protein, inhibits cell survival upon genotoxic stress [116].

Despite the relative easiness of validation through the activity, direct validation of proteomic results by this means is relatively scarce in the scientific literature. However, the few existing results often confirm that enzyme activities are altered even when the proteomic screen shows a change in one or a few spots corresponding to the enzyme, without necessarily a global change in the total amount of the protein [109,117,118,119,120,121]

Beyond these examples, an even more precise result can be reached when the enzyme activities are measured directly on the 2D gel spots after in gel renaturation, following the zymography principles [122,123,124]. Although kinetic measurements are difficult to carry out via such techniques, they allow mapping which spots bear the activity, which can be a very relevant information. As not all enzymes lend themselves to such in vitro renaturation and to gel assays, there are only a few examples in the literature [125,126,127,128,129].

Hopefully, validation of 2DGE proteomic results is not limited to enzymes, and indirect validations can be of high interest. As the name says, indirect validation does not validate directly the activity of the proteins of interest, but consequences on the function(s)/structure(s) in which these proteins play a role. One typical example is represented by the actin cytoskeleton. Actin being one of the most abundant cellular proteins, quantitative changes in the total actin amount are seldom observed. However, there are numerous proteins whose activities control the shape and dynamics of the actin cytoskeleton, and these proteins may change in their abundance/modification profile under various biological conditions. Consequently, a change in the structure of the actin cytoskeleton can be expected, and probed by confocal microscopy, as described in [130,131,132]. Detailed analysis of target proteins is also an interesting option [108,133]. Metabolic activity can also be a very good indirect validation (e.g., in [111,134,135]), and helps indeed in solving issues that are undecidable through proteomics and even enzyme activities. For example, when an increase in central metabolic enzymes is detected, does it mean that the overall energy consumption increases or that the system is trying to compensate for a defect? Measuring the glucose consumption [135] or the pyruvate level [111] solves these issues.

Another surprising benefit of 2DGE for hypothesis validation can be summarized by the motto from the architect Mies van der Rohe: “less is more”. For reasons that are linked to the analysis depth but also with some intrinsic yet not well-understood features of shotgun proteomics, this proteomic setup leads to very large lists of modulated proteins (often counting in hundreds) in differential proteomics. Such large lists are very difficult to handle by hand, so that computerized analysis of the data is the norm in shotgun proteomics. In addition to the problems caused by the variable quality of annotations, which percolate into the quality of the final results, computerized analysis is by essence collective, and the pathways that include a large number of proteins are selected over those with a low number of proteins. While normally robust, this approach leaves no room for serendipitous discoveries.

By contrast, 2DGE proteomics often yields lists of modulated proteins that count in tens instead of hundreds. Although computerized analysis can be applied to 2DGE proteomics with success (e.g., in [117,132,135]), such lists can be scrutinized for individual proteins, and sometimes a few proteins belonging to different pathways can lead to unraveling novel mechanisms. A good example is represented by pulling the thread from coactosin-like protein to cytoskeleton reorganization [132]. Proposal of mechanisms for zinc genotoxicity is also a good example where a few proteins can lead to mechanistic evidences, provided that substantial validation is carried out [131].

### 4.3. The Quest for PTMs: Unsupervised PTM Analysis as a Strength of 2DGE Proteomics

A step further in the use of the ability of 2DGE to separate modified forms of proteins resides in the identification of the modification(s) and of its (their) position(s) of the proteins. As mentioned above, taking modifications into account is one of the great strengths of proteomics over transcriptomics, for example. In order to perform this task that can be daunting, two strategies can be devised, which can be summarized as supervised vs. unsupervised search for modifications. In the supervised search, the type of modification that will be search for is defined upfront. For example, one can search for phosphorylations only, or for acetylations only, etc. There are two important advantages in such a supervised search. The first is that the search space for modified peptides, although greatly increased compared to unmodified peptides only, remains reasonable. The second and even more important advantage is that reagents able to select the class of modified peptides of interests become usable, with a great increase in the sensitivity of the approach. Such approaches have been reviewed [136,137,138,139], so that we will not enter into any further detail. Of course, the major limitation of the supervised search is that you only look at what you are allowed to look at.

Conversely, unsupervised PTM search sets no limit at the type of modifications that can be searched for, and relies on the ability of MS/MS to take into account any modification that brings a mass difference (i.e., any modification in fact). Such an approach is clearly out of reach by direct LC-MS/MS at a proteome-wide scale, because the search space becomes much too large. However, direct unsupervised identification of modifications has been instrumental in the discovery of new modifications, quite often using histones as target proteins. Histones offer the triple advantage of being very abundant in cells, bear only few sequence variations and easy to purify by acid extraction, leading to a small enough search space. It is therefore no surprise that modifications such as methylation, acetylation [140], propionylation, butyrylation [141], malonylation, succinylation [142], or crotonylation [143], have been identified first on histones in an unsupervised way and then searched for proteome-wide in a supervised way once the suitable reagents have been developed.

The histone example shows that the key is to restrict the search space by purifying the protein of interest. In this context, 2DGE can be viewed as a micropreparative tool able to separate protein modified forms, where modifications can be searched for in an unsupervised fashion by MS/MS. Thus, 2DGE has been instrumental in evidencing protein deamidation [144], but its abilities are not restricted to this modification. In this strategy, the name of the game is to separate a modified protein form, then to search for the modification(s) that may explain the observed change in the biochemical coordinates (most often the pI). It is perfectly possible to identify phosphorylations at this stage (e.g., in [108,120,145,146,147]), and sometimes phosphorylations that have been undetected by targeted phosphoproteomic approaches can be identified by this approach (e.g., in [147]). Of course deamidation can also be identified (e.g., in [148]) but less classical modifications can be detected too. For example, this combination of 2DGE and MS/MS has been instrumental for the identification of strong cysteine oxidation [149], a modification that strongly modifies protein function [150], for which no enrichment tool has been found to date but that has been described in several proteins [151,152,153,154,155]. Other modifications, such as succinylation, have been found by this approach [154]. Last, but certainly not least, extensive PTM mapping has been performed by this unsupervised approach in some cases [107,156].

Another case of non-classical PTM is represented by covalent adducts derived from toxic chemicals and modifying nucleophilic residues on proteins such as cysteine and lysine. There are cases where the chemical itself is known and electrophilic enough to react directly with amino acids, leading to possible supervised search strategies [157]. However, many other situations exist. For example, the chemical can be reactive but not lend itself to targeted approaches. More often, the chemical of interest itself is not reactive, but one of its metabolites is, which clearly complicates the situation. In such cases 2DGE proteomics can be an interesting solution to the problem, either by analyzing in detail the modified proteins to find the modification [158], or by detecting the modified protein forms via blotting and the use of an anti-hapten antibody [159,160], or using a radioactively-labelled chemical [161,162,163,164,165,166,167,168,169]. In such approaches, the precise identification of the modified peptides is often not carried out. Despite this relative lack of precision, these approaches provide the additional information of the source and gross chemical nature of the modifications, compared to classical 2DGE proteomics.

There are also intermediate situations between completely supervised searches for PTMs and completed unsupervised strategies. For example 2DGE can be used as a micropreparative tool to enrich into modified proteins prior to digestion and enrichment of modified peptides through dedicated approaches [170]. The reverse scheme, i.e., selection of modified proteins prior to analysis by 2DGE, has also been used [171], but with no precise identification of the modification sites. The same lack of precise site identifications is often observed in approaches using 2D blotting with modification specific antibodies (e.g., in [172,173]) even if now some authors make the required additional studies to map the modification sites on the target proteins [174].

Another interesting case is represented by protein carbonylation. Carbonylation of proteins can occur by a variety of mechanisms ranging from direct amino acid degradation [175,176] to conjugation to oxidized lipids [177] or unsaturated aldehydes [178]. These diverse chemistries make targeted approaches difficult. However, a popular way of identifying carbonylated proteins, without going into the details of the modified peptides, however, uses conjugation of the protein carbonyls to a hapten (usually biotin of dinitrophenyl) via the coupling of the carbonyl to a hydrazide or hydroxylamine moiety, then display of the proteins on a 2D gel, blotting and identification of the modified proteins by an anti hapten reagent (antibody or avidin). This approach has been used in a variety of biological situations (e.g., in [179,180])

### 4.4. The Case of Protein Truncation

Besides the generally taken into account case where chemical groups are grafted on the proteins, as mentioned above, an important but underestimated PTM is represented by protein truncation and cleavage. Maybe the epitome of this situation is represented by trypsin itself. After the classical activation of trypsinogen into trypsin [181], trypsin undergoes a series of self-cleavages [181], resulting into a protease with a low chymotrypsin activity [182] before final inactivation by autolysis. This simple example shows that besides regulating the half-lives of the proteins, cleavage can also alter their activity. There is therefore considerable interest in investigating protein cleavage and truncation [183], and dedicated shotgun approaches to do so have been proposed [184,185,186]. However, these N-terminomics protocols make some implicit assumptions, e.g., that the fragment containing the neo N-terminus after cleavage is stable enough to be detected, and that the neo N-terminal peptide has features (e.g., length, hydrophobicity) that make it detectable directly in LC-MS/MS. Complementary protocols looking for C-termini have been proposed [187], but are of much lower sensitivity.

By contrast, using 2DGE for the study of protein truncation makes the simple assumption that at least one of the protein fragments will fall in the separation space and will be sufficiently different from the parent protein to be separated from it. Noteworthy, 2D zymography has often given evidence for truncated proteins being enzymatically active [125,128,129]. Thus, 2DGE proteomics has been used to study protein cleavage, sometimes in a supervised way to study specifically the action of specific proteases [188,189,190], but more often in an unsupervised format to detect increased spots in the condition of interest, which are then identified as cleavage products (e.g., in [191,192]). This approach has been used with success in the food area to assess the post-mortem degradation of muscles, for example in meat [193,194] or in fish [195,196,197,198].

A summary of strengths and weaknesses of 2DGE-based and shotgun proteomics is given in Table 1.

## 5. 2DGE Proteomics in the Most Difficult Field: Clinical Proteomics

The field of clinical proteomics is huge, rewarding, but difficult, as proteomics faces the full force of human variability in this field. Indeed, biomarker discovery studies, independent from which method they use, while leading to a continuously growing number of potential candidates, most frequently originate in academic research settings. This is due to challenges in the transitioning of the results “from bench to bedside” which requires large-scale validation and clinical trials. Those studies are difficult to accomplish by a single academic lab or even institution [199] and require considerable resources. Overall, the complexity of translational research has turned out being the bottleneck for the implementation of clinical tests. Consequently, many potential markers are identified and published but require further investigation. A comprehensive overview for challenges in biomarker discovery studies is given in e.g., [200,201,202].

Indeed, in the subfield of cancer biomarker discovery, there are still some long-term success stories using 2DGE to report, although the following few selected examples are of course not a comprehensive review of the hundreds of publication in the field of 2DGE clinical proteomics:

The great potential of 2DGE in clinical research was already demonstrated by Charrier et al. in 1999 [203]. Authors were interested in the discrimination of prostate cancer (PCa) vs. benign prostate hyperplasia (BPH), which is a nonmalignant form of prostate disease. It was known that PSA (prostate specific antigen) the main marker for PCa is able to bind protease inhibitors in serum and this binding can be used for discrimination between PCa and BPH on basis of the free to total PSA ratio. Charrier and colleagues detected substantial numbers of cleaved (inactive) proteoforms of PSA being relevant for the diagnosis of PCa. An extension of the study to more than 90 patients showed that in BPH significantly more low-molecular weight PSA forms were present in serum compared to PCa allowing for a significantly improved diagnosis of PCa [204]. Those findings were only possible with top-down approaches like 2DGE. Ironically, these findings could not be translated into a classical biomarker kit because of the difficulty of the classical formats to handle proteoform and especially cleavage issues.

In addition to this example the group around Tadashi Kondo in Japan has used 2DGE in many cancer-related studies (reviewed in [205]). They investigated, e.g., esophageal cancer [206], lung cancer [207], and liver cancer [208,209,210]. For gastrointestinal stromal tumors (GIST) they could identify pfetin as a biomarker for postoperative recurrence [211]. Pfetin could be further verified in >500 cases in seven hospitals and a monoclonal antibody was developed for immunostaining and commercialization [212]. Clinical relevance and significance of pfetin has since been validated in prospective clinical trials [213].

Another example of the efforts required in validating results from 2DGE proteomics toward clinical applications is represented by the work carried out on cerebral strokes. From the first exploratory work using 2DGE proteomics [214], it took from one year to over a decade of work to reach an encouraging and relevant level of clinical validation [215,216,217,218,219].

We ourselves have worked in the field of clinical proteomics for years now. Some studies were also performed using 2DGE. The most successful studies were those conducted in close collaboration with excellent pathologists. One of the studies involved the identification and verification of a protein biomarker candidate for liver cirrhosis. Here we started with a 2DGE-based study of manually microdissected cirrhotic liver tissue from 7 patients identifying human microfibril–associated protein 4 (MFAP-4) as a potential diagnostic marker for hepatic cirrhosis [220]. In a next step, MFAP-4 could be verified as a biomarker in sera of patients with liver cirrhosis of different etiology and of different stages by Western-blotting and ELISA in >100 independent samples. MFAP-4 could be further confirmed in an extensive follow-up study 7 years later [221] focusing on its potential as biomarker for the differentiation of no to moderate (F0–F2) and severe fibrosis stages and cirrhosis. Here, MFAP-4 was verified in a retrospective study including *n* = 542 hepatitis C patients in serum using an AlphaLISA immunoassay. The clear result was that MFAP-4 in combination with existing tests leads to a more accurate non-invasive diagnosis of hepatic fibrosis and allows a cost-effective disease management in the era of new direct acting antivirals.

Complete success stories arising from proteomics in the field of clinics are difficult to report, either because the gap between what could be achieved at the academic level and what was required by the biomarker industry could not be closed (MFAP-4 example), or because the best clinical value was offered by a combination of markers, which is often felt as cost-ineffective [219], or because the relevant results offered by 2DGE proteomics at the proteoform level could not be easily transposed in a classical, user-friendly bioassay kit [204].

## 6. As a Conclusion: May Look Slow and Cumbersome, but Still Valuable If Not Irreplaceable

When trying to summarize, concisely, all the above examples, a few trends emerge on the utility of 2DGE proteomics in biological sciences. The outstanding ability of 2DGE proteomics is its ability to analyze proteoforms at low cost and with a relatively high efficiency, thereby moving away proteomics from areas where it falls in concurrence with sequencing-based transcriptomics, which is faster and much more comprehensive. The limitations of 2DGE proteomics in terms of protein hydrophobicity, molecular weight and pI extremes are well known for decades, but are also shared by MS-based top-down proteomic approaches, and are indeed linked to the physico-chemical behavior of proteins themselves. Consequently, the fact that electrophoresis is more efficient than chromatography to separate and analyze proteins (while the reverse is true for peptides) is as true now than it was at the end of the 20th century when it was published [222]. It is, therefore, no surprise that low-resolution variants of electrophoretic methods have been used as a prefractionation tool in MS-based top-down proteomics [223]. The second strength of 2DGE proteomics is its easy interface with MS, which has revolutionized the degree of detail and precision that can be reached with 2DGE proteomics, without the need of any peptide enrichment upfront (see for example [107]), but also with western blotting, which remains a very efficient tool in many proteomic studies, such as the immunome/allergome ones cited above. Moreover, a positive consequence of the fact that 2DGE proteomics takes proteoforms into account, and is, therefore, closer to the cellular physiology, and is felt downstream of the proteomic phase, when functional validation must be carried out. A perspective for future use of 2DGE would be as an enrichment tool prior to top-down MS; therefore, analyzing complete proteins eluted from the 2D gels instead of the peptides produced by in-gel digestion of the proteins. This would couple the superior separating power of 2DGE [222] with the exquisite details provided by top-down MS [224]. It must be underlined that this approach has been described in the past with some success [225,226]. However, it is less straightforward that it may seem. Protein elution from 2D gels require SDS (sodium-dodecyl-sulfate), which must be removed prior to the top-down MS [227], with serious risks of protein losses. Furthermore, an often neglected problem lies in the oxidative modifications of the proteins that can be brought by the electrophoretic process itself [228], which will complicate the downstream top-down analysis and requires special precautions to limit this artifact [228,229].

Nevertheless, and overall, the tradeoff between 2DGE proteomics and shotgun proteomics exchanges details and precision against speed and analysis depth in terms of number of gene products, and this is a choice to be seriously considered beyond fashion.

## Figures and Tables

**Figure 1 proteomes-08-00017-f001:**
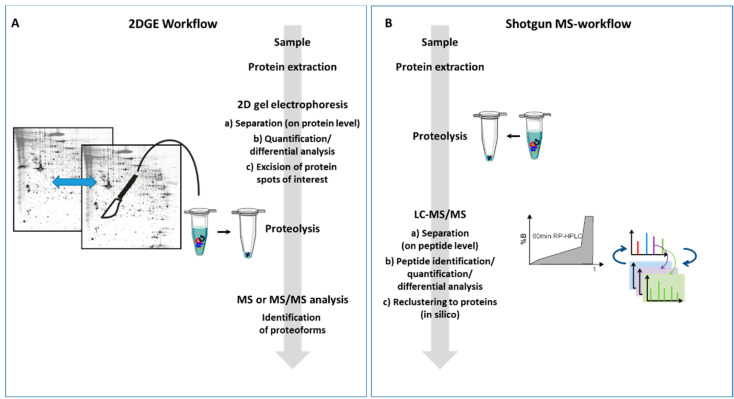
Comparison of two-dimensional gel electrophoresis (2DGE) and shotgun MS-workflow. The main differences between the two workflows are the type of molecule separation and the method of quantification. While in the 2DGE approach, (**A**) the proteins are already separated; in the shotgun approach, (**B**) the proteins are first digested and then the resulting peptides are separated. In (**A**) proteins or proteoforms are detected, quantified, and identified, whereas in (**B**) the detection, identification and quantification is performed at the peptide level. The peptide data are then used for protein reclustering.

**Figure 2 proteomes-08-00017-f002:**
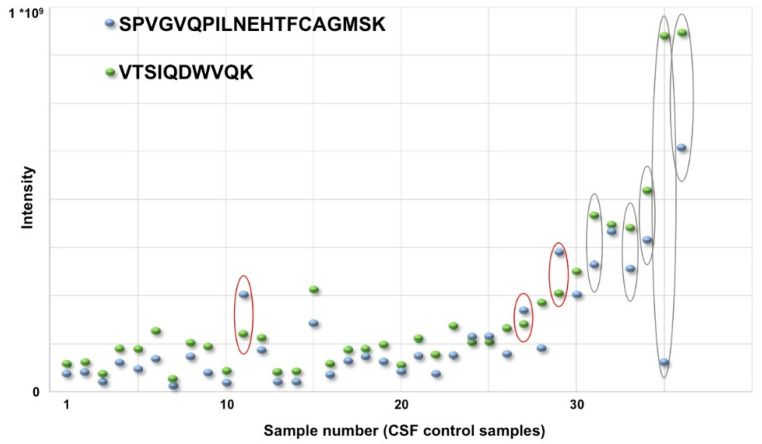
Variability of haptoglobin peptides in cerebrospinal fluid (CSF). Two tryptic peptides (SPVGVQPILNEHTFCAGMSK, black, and VTSIQDWVQK, green) were selected as representatives for haptoglobin. Both peptides were detected in 36 CSF samples of healthy control subjects. In samples 11, 27, and 29 (marked with a red circle) the blue peptide was more intense than the green peptide. In all other samples, the intensity of the green peptide was similar or higher than the intensity of the blue peptide. Additionally, the difference in intensity of both peptides showed high variability in at least some of the samples (e.g., 31, 33–35, and 36, marked with a black circle). Summarized, the variability of intensity of some of the unique tryptic haptoglobin peptides has made a valid protein quantification on basis of the peptide amounts in this study cohort impossible.

**Table 1 proteomes-08-00017-t001:** Strengths and weaknesses of 2DGE proteomics vs. shotgun proteomics.

	2D Gel-based Proteomics	Shotgun Proteomics
**Sample consuming**	++(+) *	+
**Time consuming**	+++	++
**Analysis depth**	++	+++
**Separation/identification**		
Separation/detection of proteoforms	+++	+

Identification on protein level	Multiple identifications	Only by inference
		from peptides
Detection of proteoforms	+++	-
Details at peptide level (e.g., sequence coverage)	+++	+
Number of modulated proteins identified	+	+++
**Coupling with biochemical methods**		
Antibodies	+++	+
Enzymes (zymography)	+	-
**Robustness of quantification**		
Sensitivity	++	+++
Linearity	+++	+
**Need of validation**	+++	+++

* depending on 2D gel-based technique used.

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
