# Peer review of "What Room for Two-Dimensional Gel-Based Proteomics in a Shotgun Proteomics World?"

_proteomes, 2020, doi:10.3390/proteomes8030017_

Round 1

Reviewer 1 Report

I read with great interest a new review by Rabilloud and colleagues: “What room for two-dimensional gel-based proteomics in a shotgun proteomics world ?” by Marcus, Lelong and Rabilloud. I was not disappointed. This is another timely, critical, and insightful look at 2D gel electrophoresis and indeed the field of proteomics as a whole. I do have a few suggestions, comments, and corrections that I hope the authors accept with the best intentions they are intended. In some cases I seek to clarify or refine English language usage as I have in the past had occasional comments from students   -  who love that I use the work of Rabilloud and colleagues in my teaching  -  as to the exact intent of what was being said. Rather than impose my own interpretation in such instances (which I would hope, to date, has been correct more often than not), for this review I will try and clarify those instances that I think might fit this particular issue (respectfully noting that if I had to write this in French or German, the manuscript would not exist).

Major:

Line 16: “Although this opinion may be completely true for some biological questions”  -  somewhat vague or open-ended; would suggest being specific or dropping this.

Line 18: Would ‘proteoforms’ be a better word than “isoforms”?

Line 42: What exactly is meant by “…especially when the mechanistic details of the cellular responses are not the core of the biological question”?

Line 49: Perhaps ‘less than optimal’ might be less confrontational than “the worst possible setup”?

Line 53: With reference to peptide mass fingerprinting is there any comparable data for tandem MS? Certainly, in our experience, when we send excised digested spots to an MS lab/facility that mostly does shotgun work, they consistently comment that they get better data overall from the gel-resolved proteins.

Lines 55-56: The first report of third separations from 2D gels (i.e. 3DE) was almost a decade earlier (Postfractionation for Enhanced Proteomic Analyses: Routine Electrophoretic Methods Increase the Resolution of Standard 2D-PAGE. J. Proteome Res., 4, 982-991, 2005), and the method has been utilized in other studies since. Although this work was not cited by Colignon et al., their method is certainly an effective complement to the original method.

Lines 63-64: I think I understand what the sentence is trying to convey, but could it please be rewritten to ensure clarity?

Lines 65-67: (i) Perhaps rephrase as  ‘…the extent of this theoretical advantage…’? (ii) I think the work of Thiede et al., and Zhan et al., among others (including our own more recent, as yet largely unpublished, datasets), quantitatively obviates this notion that the shotgun approach has a larger analytical window relative to 2DGE. If anything, it is arguably smaller. (iii) Actually, if one applies the Scientific Method and expects samples to be minimally analyzed as technical triplicates, then 2DGE (enabling parallel sample resolution) is likely only marginally slower overall relative to shotgun; doing the calculations based on current ‘best’ routine shotgun analytical times per sample, with technical replicates (which are seemingly often not carried out!?), suggests that not all that much is gained in terms of throughput times  -  and of course, there is all the critical information that is lost to consider as well.  Thoughts?

Lines 73 & 90: the term ‘reassembled’ is used. Although more objective detail is indeed given later in the text, while I suspect unintended, I personally find this wording just a little misleading because I read it and think ‘complete protein sequence’ (e.g. you don’t talk about reassembling a watch or a car and expect an incomplete entity). More often than not, as pointed out later, protein identifications are inferred, often from only a ‘few’ peptides. Indeed, it is common and agreed practice among mass spectrometry firms to talk about the advantages of their latest products by noting that ‘so-and-so many more proteins were identified’ when many so-called identifications are based on only one or two peptides…….and this is seem in the scientific literature itself as well. Thus, perhaps the concept of ‘inference’ should already be introduced at this earlier point?

Lines 139-140: “Quantification at [the] peptide level alone might be a possible solution to this problem.” Could this statement please be expanded on for clarity?

Line 229: “..proteomics lacks comprehensiveness..”  -  a rather bold blanket statement without associated criteria, that essentially comes out of nowhere. Can this perhaps be put into some context? As a counterpoint, I would suggest transcriptomics lacks comprehensiveness because, like genomics, the data tell us nothing about PTM (although do inform about splice variants) and thus potentially active species, nor even about how much of the resulting amino acid sequence is actually present. Thoughts?

Line 258: “…better than nothing…” In English, this may be interpreted as rather flippant considering the overall sentiment of the manuscript. Was this the intention? If not, perhaps ‘…better than other available approaches/techniques…’?

Lines 309-310: “…seen only as a hypothesis generator…” I would argue it is a method that both tests and generates hypotheses  -  as it true of any good scientific approach. Every good experiment leads to new/better questions.

Line 545: “slow, cumbersome”  Sorry, but I find this somewhat hackneyed (see above in terms of fast/slow as well). I feel it necessary to note that there is nothing ‘free and easy’ about doing LC/MS/MS (i.e. the core of shotgun), especially when there are problems (as so often seem to occur) with either the front end or the mass spectrometer. Would the authors be comfortable dropping these adjectives, recognized that they are perhaps expressing somewhat personal feelings? From the same sort of personal perspective I would have to say that even if these descriptors were broadly true, I’d rather be getting things ‘right’ than relying only on incomplete amino acid backbones.

Line 550-551: As above. I am quite surprized to see this dogmatic statement that has been repetitively copied from 30-40 year old review articles, and most often by folks promulgating the shotgun approach who are (purposefully?) unaware of advances in 2DE over the last 20+ years. Do the authors really want to perpetuate this rather outdated ‘quote’? I believe I understand the sentiment but is this the best possible statement to make the point………….or is it actually even fully accurate anymore for that matter?

Line 564: “analysis depth”?  As noted in comments above (i.e. concerns Lines 65-67) I believe it more than fair to say that published data now quite effectively demonstrate that 2DGE/LC/MS/MS provides greater analytical depth than does the shotgun approach. The real question is whether that proteome depth needs to be routinely analyzed or only when ‘full’ proteome coverage is sought? With this in mind, it is also important to note that ‘depth’ as used in the manuscript seems only to refer to amino acid sequences (and not even full coverage of those) rather than actual proteome coverage in terms of protein species.

Minor:

Line 28: 1970’s

Line 29: “Since [then]…”

Line 103: “is’ should be ‘are’

Line 117: should sentence ending there with “….protein.” perhaps say ‘…..the protein and its variants’?

Line 130” “…as [a] potential…”

Fig 1: Could the black data points perhaps be made more intense to better distinguish them from the green?

Line 156: Might  “Before, already several studies have identified…” be rephrased as ‘Previously, several studies already identified…’?

Line 159: Might “…for the distinction of differential proteoform…” be rephrased as “…for the differential detection/identification of proteoforms…”?

Line 162: Might ‘design’ be a better word than “construction”?

Line 196: Might ‘subsequent’ be a better word than “following”?

Line 211: Should the sentence ending there with “background” be referenced?

Line 213: Might ‘target’ be a better word than “point”?

Line 226: ‘antibody’ rather than “antibodies”.

Lines 326-327: “This should be come as a surprise…” should read ‘This should come as no surprise…’

Line 365: “which” should be ‘whose’ or ‘the activities of which...’

Line 501: ‘filed’ for “filed”

Author Response

Dear Editor and reviewers

First and foremost, we would like to thank you very warmly for the time and effort that you have devoted to our manuscript. Your comments were found to be very valuable to improve the manuscript. In order to ease the assessment of the revision process we have adopted a color code for the changes made to the manuscript.

Pink corresponds to changes made according to reviewer's 1 comments

Blue corresponds to changes made according to reviewer's 2 comments

Orange corresponds to changes made according to reviewer's 3 comments

Regarding reviewer's 4 comments, we have added the figure (now Figure 1) and Table requested

Before we detail the changes made, we would like to address an important point risen by reviewers 1 and 3. In their reports, they insist on a strategy aiming at identifying as many proteins as possible from a single 2D spot, using ESI/MS-MS as both an identification and a quantification tool.
We agree that this strategy looks extremely exciting, but we have identified some caveats that render us very cautious about its real practicability, especially for differential proteomics as is most often practiced.

The first, and minor, caveat resides in the quantification of the MS data themselves, and especially in their renormalization. In shotgun proteomics, the data for individual peptides are renormalized against the sum of all peptide data, using the fact that most protein abundances do not change upon any non lethal perturbation of the biological system. This constant background will be absent in analyses from 2D gel spots, so that the renormalization of the data is far from trivial. We have begun to touch such issues in a previous paper of ours (Prudent et al. doi 10.1371/journal.pone.0208979) and found this issue not to be obvious to solve

The second, and more important, caveat lies in how to apply this ESI/MS-MS for differential proteomics. What the published work using this strategy tells us is that what we call a spot is in fact the convolution of several gaussian protein peaks of various abundances. In other words a single spot is the sum of several protein components P1…Pn.
The staining pattern should be seen as a horizontal cross section of this multipeak massif at a defined height, driven by the sensitivity of the stain and the loaded amount. If we trust MS indices such as emPAI (which is clearly arguable), there are major and minor contributors to the total, observed spots.

Then what happens if we want to use this approach to see what changes between two different situations and thus between two different 2D gels. We will excise homologous spots in gel A and gel B, analyze both by ESI/MS-MS, and use the MS signals to derive what changes in spot X between gel A and gel B. As we all know, of course spot X will not be at the exact same position (from gel edges) in gel A and gel B, as there are always some minor migration changes from one gel to another.

However, the implicit assumption made for using this strategy is that the migration of all components making a spot is altered exactly the same way between gel A and gel B. If this is not the case, then there is a major opening for artefactual "discoveries" in which a change will be detected in component i of spot X between A and B not because there is an actual change in component i but because component i is not positioned in the same relative position compared to the major components making spot X. In other words, we see a change just because we have not cut the same portion of the gaussian distribution of component i in gels A and B.

Within this frame, it is a matter of personal opinion to decide whether the risk is worth taking. In our opinion it is not, and as commonly said, better safe than sorry.

We would also like to emphasize at this point that the risk increases when the separation performances of the 2D decrease, e.g. at high protein loads. Under such conditions two problems are encountered:

i) spots have a wider gaussian spread. This means that neighboring spots interfere more, and this will be an even greater problem when vicinal spots begin not to move exactly the same way between different gels.

ii) spots begin to elongate then streak in a rather unpredictable way indeed.

For abundant components of a spot we can figure out what is going on via the staining pattern, but for more minor components that marginally contribute to the staining but are detected by the MS/MS, we have no way of knowing before it is too late.

This means in turn that the only safe way to use this approach is via a coupled silac/comigration approach, as described in the 2012 Thiede et al MCP paper. This however carries over the limitations of the silac approach such as low multiplexing (same as DIGE indeed), applicability to a very limited range of samples (mostly cell culture) and perturbations to the cellular physiology and thus to the validity of the biology brought by the peculiar cell culture conditions imposed by the silac labelling.

Thus although theoretically quite interesting (although controversial) we feel that such a topic is not suited for the review as we want to frame it, and would prefer not to enter this game in this paper.

We would like then to switch to the detail of our responses to reviewers.
_______________________________________

Reviewer 1, Major comments

Line 16 in the abstract: we propose a new wording for the sentence
Line 18:

change made

Line 42 (now lines 43-44) details added

line 49 (now line 50) we have kept the worst possible setup, but added a precision to make it less confrontational

former line 53 : we do agree that because of the prepurification brought by the 2DGE separation, the MS data afterwards are much cleaner and better than in a shotgun soup. However, we do not think that such details are in the frame of the paper

former lines 55-56 (now line 63) reference added.

former lines 63-64 (now lines 67-71) we have added an example to clarify the message.

former lines 73&90 (now lines 83-84&111). We have changed "reassembled" to "reclustered" to take the point of completeness implied in the word "reassembled"

former lines 139-140 now lines 162-163: we have added a sentence to clarify the message

former lines 229, comment on comprehensiveness. We have added a long sentence (lines 259-262) to clarify what we meant by this statement and where it rooted from

former line 258 "better than nothing". Changed to be less flippant (lines 290-292)

former lines 309-310 "hypothesis generator". Wording changed (line 344)

former line 545 "slow cumbersome" wording changed (line 590)

former lines 550-551: we are sorry to disagree with the reviewer, but there are intrinsic limitations to the 2DGE separations, which we tried to detail more precisely in lines 595-596.

Comments on analysis depth have been addressed earlier in this rebuttal letter, and all minor comments have been taken into account.
_________________________________________

Reviewer 2

former line 82 (now line 106) wording changed according to reviewer's suggestion

Figure (now figure 2) Black changed to blue

comment on 2D blot/gel realignment (former lines 214-215) sentences and references added to take the point risen by the reviewer into account (now lines 239-245)

comments on autoantibodies: taken into account with references added on lines 253-254
__________________________________________

Reviewer 3

comment 1; changed as suggested

comment 2 : in the Hoogland paper, at that time eukaryots amounted to at least 3/4 of the maps, including body fluids that are very heavily modified. This is why we keep the 3 spots/protein (on average) figure. The rest of the comment has been dealt with earlier in this rebuttal letter, and also explains why we do not wish to cite paper going into that direction.

Comment 3: we kept 2DGE throughout and explain indirectly why in lines 600-602. A list of abbreviations has been added (in red)

Comments 4-6: dealt with earlier in this rebuttal letter, and in lines 54-58

Comment 7 now addressed in lines 88-90

Comments 8 &11: much too specific for such a review, far western is just a subtype of western (protein-based) blotting (comment 8) and incorporating DNA instead of another substrate does not change the zymographic principle.

Comment 9 : with the ease of MS analyses nowadays and the fact that it will be anyhow requested by any decent reviewer of a 2DGE proteomic paper, we do no think that image databases have any practical value any longer

Comment 10 : once again the point is exactly that no change in total abundance (shotgun) does not necessarily means no change in activity, by modulation via PTMs. By the way GAPDH is probably a very bad idea as a loading control for blots, as it is the epitome of moonlighting protein (see also the papers on deja-vu by Petrak and Merriman). So the paradox is here for real. This said one very interesting feature of Martins de Souza's work is that 2D sees an increase of glycolytic enzymes. Thus it can correspond to 2 opposite real phenotypes. Either the "diseased" cells are on metabolic overdrive or oppositely they are trying to compensate for a metabolic defect. By measuring the end product of the pathway (pyruvate) Martins de Souza et al. showed that the second hypothesis is correct, and not the more straightforward first one.

Comment 12: we do maintain what we wrote. It is not the too much information that is bad, but the way it is handled nowadays. Furthermore a lot of good information is never too much, but a lot of information of variable quality (to say the least) can be of poor usability, if not misleading.
In order to show to what nonsense excess poorly handled information can lead, we suggest the following paper: Tuomela et al DOI: 10.1371/journal.pone.0068415. It is a transcriptomic paper so outside the frame of our manuscript, thus not cited. It speaks about muscle contraction in macrophages. Since when have immune cell muscles ?

As to suggested references, we believe that in a review it is the authors' privilege to select what they prefer to cite. We have just selected the Thiede et al reference for reasons explained earlier in this rebuttal letter.
_____________________________________

Reviewer 4

Figure and Table (respectively now Figure 1 and table 1) added.

Reviewer 2 Report

In their review article entitled „What room for two-dimensional gel-based proteomics in a shotgun proteomics world ?, Marcus, Rabilloud et al. – two renowned figures in the field – discuss the pros and cons of 2DGE as top-down approach, nicely covering all aspects from the basic principles to clinical applications and critically contrasting them with bottom-up shotgun approaches. The manuscript is comprehensive (also by literature cited) and well written, and will provide an important resource particularly to younger proteomic researchers who were likely not grown up with own hands-on experience on 2DGE.

I have only a few minor remarks and some suggestions for literature still worth to be considered because of fit to the manuscript and/or impact to particular research fields.

P. 2, l. 82-85: as the authors introduce the term “peptide ion mass spectra” for MS1 spectra, they should be consistent and use the term “fragment ion mass spectra” for MS/MS or MS2 spectra.

Fig. 1: the black symbols appear blue, at least in the PDF sent out for revision. Legend: there are no green or black peptides, please rephrase.

P. 5, l. 222-223: it is a bit unfortunate that the authors are very concise on autoantigen identification. Such an important aspect would merit a few more sentences and somewhat broader citations. References provided are mainly centered around arthritis, neglecting the impact that 2D immunoblotting followed by decoration with patient antisera had in Multiple Sclerosis research (e.g. PMIDs 17846150, 19416878).

Along similar lines, the authors describe one central part of this approach simply as “… looping back to host protein 2D gels to identify then the proteins of interest …” (p. 5, l. 214-215). However, this “looping back” from a 2D immunoblot to a 2D gel for mass spectrometric protein identification, which is also of key relevance for the supervised PTM analysis described later in the manuscript, is not as trivial as it sounds. It would thus be helpful for the reader if the authors could expand a bit on the challenges of such parallel gel approaches for PTM detection and recent work in this journal (PMID 28248254) together with references therein may be a good starting point.

Author Response

(The authors gave the same response as above.)

Reviewer 3 Report

This review manuscript is about two-dimensional gel electrophoresis and proteomics. In particular, the authors describe, where and why two-dimensional gel electrophoresis-based proteomics can be profitably used. The main issue is that in this review the authors are not paying attention to some very important achievements in 2DE methodology. The authors consider 2DE and shot-gun proteomics as two different competitive methods, but the point is that now they should be considered as a united approach - 2DE- based shotgun proteomics (2DE with following ESI LC-MS/MS). A classical old-fashion 2DE based on spot staining and spot density evaluation is not enough. These aspects are disclosed in details in (2) Naryzhny S.N. Towards the Full Realization of 2DE Power. Review. Proteomes 2016, 4(4), 33. (5). Naryzhny, S. Inventory of proteoforms as a current challenge of proteomics: Some technical aspects. J. Proteomics 2019, 191, 22-28) and in some other papers. The list of recommended references is in the bottom.

Here, some particular comments.

  1. The term “proteoforms” for any form of a protein is preferable. In the text, “isoform” is also used (on p.17-130 – correctly, but on p.18 – proteoform (not isoform) is better)
  2. “3 spots per protein in eukaryotes” – it is a wrong statement. This number came for all objects in 2DE database (SWISS-2DPAGE) including Arabidopsis thaliana, Dictyostelium discoideum, E.coli…(not only for eukaryotes). Actually the number for eukaryotes is higher. See (8) Thiede, et al. 2012.
  3. For two dimensional electrophoresis authors use both - 2DE and 2DGE. For consistency should be only one, and it is 2DE (as it is used in Proteomes). Also, it should be a list of abbreviations in the paper.
  4. P51-52 “… the most abundant protein in a spot, as detected by mass spectrometry, most often accounts for >75% of the total signal [12]”. The key word here “most often”, as it can be very different representation of proteins in a spot. It can be an origin of incorrect quantitation and mistakes using old-fashion spot abundance estimation. As it was estimated in (8) Thiede, et al. 2012 – “… In more than 30% of all the 2DE spots, the dominant protein accounted for less than 70% of the total protein intensity”
  5. P55-56. The authors should be more critical. The example “… should a doubt persist; a third separation can always be attempted to decipher more precisely the protein content of a spot of interest [14]. Not very good example. This is a very unpractical approach, having in hands ESI LC-MS/MS technology that allows to decipher a protein content in a spot.
  6. P162-168. Here, the authors claim that protein quantitation using spot density evaluation is better than MS-quantitation using peptides. Again, there is an issue about different proteins and their ratio in a single spot. Depending on the gel resolution, the spots often contain more than a single protein, especially in case of mammalian cells.
  7. The concept of proteoforms should be addressed in more details There is a need in more details that describes new realities of 2DE in the paragraph 4. Going to the essence of proteomics: proteoforms and post-translational modifications
  8. (10). Also, Far-Western can be also mentioned as a method of direct visualization of protein-protein interactions after 2DE.
  9. 2DE-based databases of proteins (World 2DPAGE database) are not mentioned as 2DE application. Inventory of proteins and proteoforms is a main aim of proteomics.
  10. P318-341. Here, a long discussion about comparison of 2DE and shot-gun mass spectrometry, using paper of de Souza et al [97] is presented. It was claimed that “…nine out of the ten differences that they found in 2DGE proteomics were not found by the shotgun proteomics screen”. Actually, in the original paper of de Souza et al, there is no answer, which data is true, MS or 2DE. For me, it seems that shot-gun based conclusion is correct (no differences), keeping in mind that one protein, which level is changed two times (as estimated by spot density) is G3P that is very often used as a loading control in Western blotting analysis. The final correct estimation could be done by 2DE with following ESI LC-MS/MS. This discussion should be added to this part.
  11. P354-356. If the authors decided to describe approaches of enzyme detection after 2DE, more approaches on study enzymes using 2DE can be presented. For instance, detection of enzymes/proteins that are using incorporated in the gel DNA (in-gel activity) not only zymograms (10)
  12. P376-392. This paragraph should be critically rewritten. Here, a reader (especially new in this area) will get wrong impression that too much information produced by MS is not good for analysis. But the point is how to extract the necessary information from the big volume of data.

Some references to cite

  1. Klose J From 2-D electrophoresis to proteomics. Electrophoresis. 2009. PMID: 19517494
  2. Naryzhny S.N. Towards the Full Realization of 2DE Power. Review. Proteomes 2016, 4(4), 33;
  3. S.J. Fey, P.M. Larsen 2D or not 2D. Two-dimensional gel electrophoresis Curr Opin Chem Biol, 5 (1) (February 2001), pp. 26-33
  4. Lee PY, Saraygord-Afshari N, Low TY. The evolution of two-dimensional gel electrophoresis - from proteomics to emerging alternative applications. J Chromatogr A. 2020 Mar 29;1615:460763.
  5. Naryzhny, S. Inventory of proteoforms as a current challenge of proteomics: Some technical aspects. J. Proteomics 2019, 191, 22-28
  6. Naryzhny S, Klopov N, Ronzhina N, Zorina E, Zgoda V, Kleyst O, Belyakova N, Legina O. A database for inventory of proteoform profiles: "2DE-pattern". Electrophoresis. 2020 Jun;41(12):1118-1124.
  7. Thiede, et al. High Resolution Quantitative Proteomics of HeLa Cells Protein Species Using Stable Isotope Labeling with Amino Acids in Cell Culture(SILAC), Two-Dimensional Gel Electrophoresis(2DE) and Nano-Liquid Chromatograpohy Coupled to an LTQ-OrbitrapMass Spectrometer. Mol. Cell. Proteom. 2012, 12, 529–538.

Author Response

(The authors gave the same response as above.)

Reviewer 4 Report

In the manuscript, the author comprehensively described the technical advantages and disadvantages of two-dimensional gel-based proteomics compared with shotgun proteomics. The mechanisms underlying the protocols and procedures for proteomic analyses by two-dimensional gel were clearly elucidated and most updated literature was also provided in this review. Basically, the text part is well organized and well written. The reviewer highly recommends the publication of this manuscript after making following minor revisions

  • The author should give a brief introduction to knowledge and technical principle of shot gun proteomics. It is better to give a scheme or figure to compare the workflow of two-dimensional gel-based proteomics and shotgun proteomics.
  • A table summarizing the strengths of two-dimensional gel-based proteomics compared with shotgun proteomics (e.g. unsupervised PTM analysis to separate different modified protein forms vs LC-MS/MS in search for supervised protein forms; diversity of validation methods of two-dimensional gel-based proteomics using antibodies or enzymes) should be provided.

Author Response

(The authors gave the same response as above.)

Round 2

Reviewer 3 Report

Dear authors,

I respect and appreciate all the efforts consumed in the generation of the manuscript. As the title says, the authors are trying to find for 2DE a place in the shotgun proteomics. But in the text, the authors constantly oppose classical 2DE-based proteomics to proteomics based on shotgun mass-spectrometry. It looks like they are fighting the windmills. We all know the positive and weak sides of 2DE, as well as the weak sides of shotgun mass-spectrometry. And only a combination (not opposition) of 2DE with shotgun mass-spectrometry (ESI LC-MS/MS) gives 2DE methodology a new life. I was talking about this in the previous review, pointing to the recent publications. But the authors remain unconvinced and do not admit even evident things. Especially it concerns the differential analysis of 2DE images. Here, the authors claim that it is enough to quantitate the major proteins by spot density. I am sorry, but I need to repeat that the important point is about the percentage of all proteins (proteoforms) in a spot. Though the peptide-based analysis has not high accuracy, there is no alternative here. In a classical 2DE, measuring the spot density it is assumed that all this density belongs to a single protein. It is not true for many 2DE spots of the mammalian cell extracts. For instance, it can be two most abundant proteoforms, or more. At this situation, ESI LC-MS/MS of these spots can give us information about relative input of each proteoform inside a spot, it is real density. This is a way how 2DE- ESI LC-MS/MS can work in the differential analysis.

Still no response was received for some comments.

  1. Line 39-40. …With an average of 3 spots per protein in eukaryotes 39 [8], this corresponds to ca. 800 gene products. But in [8] …4000 identified spots corresponding to 1200 different protein entries in 36 reference maps SWISS-2DPAGE from human, mouse, Arabidopsis thaliana, Dictyostelium discoideum, Escherichia coli, Saccharomyces cerevisiae and Staphylococcus aureus origins (not only eukaryotes).   Though in the comment 2, the authors say: in the Hoogland paper, at that time eukaryotes (again!) amounted to at least 3/4 of the maps, including body fluids that are very heavily modified…
  2. Line 53-58. The authors say   “…paper had described that in 30% of the cases, the most abundant protein accounted for less than 70% of the “protein intensity” [13]. It should be emphasized, however, that mass spectrometry-based indexes such as emPAI, which have been shown to lack accuracy [14]”. Yes, it’s true about accuracy of the absolute protein concentration measurement. If we are estimating the relative presentation of different proteins in a sample (spot) it is not a bad choice. And if there is underestimation of some proteins abundance [14], this is for the minor components [14]. Thus, the real input of the most abundant protein in the spot density can be even less.

  1. Line 145-150   A discrepancy here -   “In a CSF (cerebrospinal fluid) study [38], which we partially published in 2018… “   Citation [38]- “Protein variability in cerebrospinal fluid and its possible implications for neurological protein biomarker research. Schilde, L. et al. PLOS ONE 2018, 13” (It’s a publication of the German group not the authors’. And this paper is about high protein variability (especially Hp) in CSF, not about Haptoglobin proteoforms). It seems that the data presented in Fig.2 were not published before and should be critically analyzed (or removed) not be just as an example of impossibility of usage of unique tryptic haptoglobin peptides for protein quantification. Keeping in mind high variability of Hp, there is a possibility (though rare) of variation of beta-chain in some patients (35, 36, for instance).
  2. Line 209-216. “One example of this case is represented by the analysis of therapeutic protein batches, where 2DGE has been successfully applied [56,57].” It is a good example of the protein quantitation using 2DE of a purified sample, but not for a whole mammalian extract. See comments above.
  3. Line 280-282. “…Consequently, most proteins appear not as a single spot, but as a trail of spots, with an average of 3 spots/protein in mammalian cellular proteins [8].” In [8], there is no numbers for mammalian cells. Same mistake as it was mentioned   in 2. Accordingly, talking only about mammalian cellular proteins should be another reference used (for instance [13] instead of [8]) and added “at least 3”.
  4. Comment to the answer to the comment 12: Sorry, but in paper Tuomela et al, there is nothing about muscle contraction in macrophages. May be it should be another reference. A data quality is an important issue, but a lot of information produced by MS does not automatically means that it will be badly managed. The same situation can be happening with 2DE data.
